# Uncovering Meanings of Embeddings
# via Partial Orthogonality

Yibo Jiang[1], Bryon Aragam[2], and Victor Veitch[3,4]

[1]*Department of Computer Science, University of Chicago*
[2]*Booth School of Business, University of Chicago*
[3]*Department of Statistics, University of Chicago*
[4]*Data Science Institute, University of Chicago*

## Abstract

Machine learning tools often rely on embedding text as vectors of real numbers. In this paper, we study how the *semantic* structure of language is encoded in the *algebraic* structure of such embeddings. Specifically, we look at a notion of "semantic independence" capturing the idea that, e.g., "eggplant" and "tomato" are independent given "vegetable". Although such examples are intuitive, it is difficult to formalize such a notion of semantic independence. The key observation here is that any sensible formalization should obey a set of so-called independence axioms, and thus any algebraic encoding of this structure should also obey these axioms. This leads us naturally to use *partial orthogonality* as the relevant algebraic structure. We develop theory and methods that allow us to demonstrate that partial orthogonality does indeed capture semantic independence. Complementary to this, we also introduce the concept of *independence preserving embeddings* where embeddings preserve the conditional independence structures of a distribution, and we prove the existence of such embeddings and approximations to them.

## 1 Introduction

This paper concerns the question of how semantic meaning is encoded in neural embeddings, such as those produced by [Rad+21]. There is strong empirical evidence that these embeddings—vectors of real numbers—capture the semantic meaning of the underlying text. For example, classical results show that word embeddings can be used for analogical reasoning [e.g., Mik+13; PSM14], and such embeddings are the backbone of modern generative AI systems [e.g., Ram+22; Bub+23; Sah+22; Dev+18]. The high-level question we're interested in is: *How is the **semantic** structure of text encoded in the **algebraic** structure of embeddings?* In this paper, we provide evidence that the concept of *partial orthogonality* plays a key role.

The first step is to identify the semantic structure of interest. Intuitively, words or phrases possess a notion of semantic independence, which does not have to be statistical in nature. For example, the word "eggplant" seems more similar to "tomato" than to "ennui". Yet, if we were to "condition" on the common property of "vegetable", then "eggplant" and "tomato" should be "independent". And, if we condition on both "vegetable" and "purple", then "eggplant" may be "independent" of all other words. However, it is difficult to formalize what is meant by "independent" and "condition on" in these informal statements. Accordingly, it is hard to establish a formal definition of semantic independence, and thus it is challenging to explore how this structure might be encoded algebraically!

The key observation in this paper is to recall that most reasonable concepts of "independence" adhere to a common set of axioms similar to those defining probabilistic conditional independence. Formally, this abstract idea is captured by the axioms of the so-called *independence models* [Lau96]. Thus, if

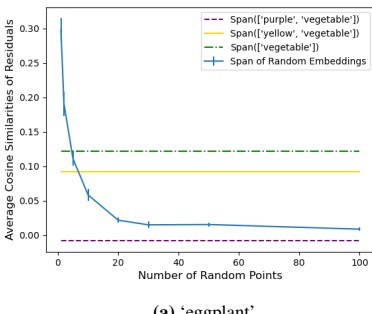

**(a)** 'eggplant'

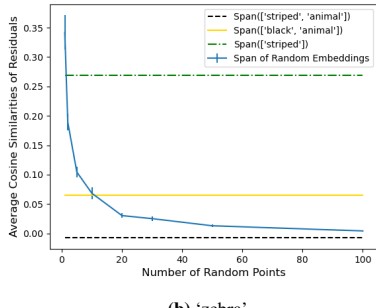

**(b)** 'zebra'

**Figure 1:** For target embedding of "eggplant", the set of embeddings that include "purple" and "vegetable" forms the subspace such that after projection, the residual of 'eggplant' has the lowest cosine similarity with residuals of other test embeddings. This matches our intuition for the meaning of "eggplant". Similarly, for target embedding of "zebra", the set of embeddings that include "striped" and "animal" forms the most suitable subspace.

semantic independence is encoded algebraically, it should be encoded as an algebraic structure that respects these axioms. In this paper, we use a natural candidate independence model in vector spaces known as *partial orthogonality* [Lau96; AAZ22]. Here, for two vectors $v_a$ and $v_b$ and a conditioning set of vectors $v_C$, partial orthogonality takes $v_a$ independent $v_b$ given $v_C$ if the residuals of $v_a$ and $v_b$ are orthogonal after projecting onto the span of $v_C$. *We discover that this particular tool is indeed valuable for understanding CLIP embeddings.* For instance, Figure 1 shows that after projecting onto the linear subspace spanned by CLIP embeddings of "purple" and "vegetable", the residual of embedding "eggplant" has on average low cosine similarity with the residuals of random test embeddings, which also matches our intuitive understanding of the word.

Since partial orthogonality is an independence model, we can go one step further to define *Markov boundaries* for embeddings as well. Drawing inspiration from graphical models, it is reasonable to expect that the Markov boundary of any target embedding should constitute a minimal collection of embeddings that encompasses valuable information regarding the target. Unlike classical applications of partial orthogonality in regression and Gaussian models, however, the geometry of embeddings presents several subtle technical challenges to directly adopting the usual notion of Markov boundary. First, the *intersection axiom* never holds for practical embeddings, which makes the standard Markov boundary non-unique. More importantly, practical embeddings could potentially incorporate distortion, noise and undergo phenomena resembling superposition [Elh+22]. Therefore, in this paper, we introduce *generalized Markov boundaries* for studying the structure of text embeddings.

**Contributions**   Specifically, we make the following contributions:

1. We adapt ideas from graphical independence models to specify the structure that should be satisfied by semantic independence. We discover that partial orthogonality in the embedding space offers a natural way of encoding semantic independence structure (Section 2).

2. We study the semantic structure of partial orthogonality via Markov boundaries. Due to the unique characteristics of embeddings and noise in learning, exact orthogonality is unlikely to hold. So, we give a distributional relaxation of the Markov boundary and use this to provide a practical algorithm for finding generalized Markov boundaries and measuring the semantic independence induced by generalized Markov boundaries (Section 3.2).

3. We introduce the concept of *independence preserving embeddings*, which studies how embeddings can be used to maintain the independence structure of distributions. This holds its own intrigue for further research (Section 4).

4. Finally, we design and conduct experimental evaluations on CLIP text embeddings, finding that the partial orthogonality structure and generalized Markov boundary encode semantic structure (Section 5).

Throughout, we use CLIP text embeddings as a running example, though the method and theory presented can be applied more broadly.

**Related work** There are many papers [e.g., Aro+16; GAM17; AH19; EDH19; Tra+23; Per+23; Lee+23; MEP23; Wan+23] connecting semantic meanings and algebraic structures of popular embeddings like CLIP [Rad+21], Glove [PSM14] and word2vec [Mik+13]. Simple arithmetic on these embeddings reveals that they carry semantic meanings. The most popular arithmetic operation is called linear analogy [EDH19]. There are several papers trying to understand the reasoning behind this phenomenon. Arora et al. [Aro+16] explains this by proposing the latent variable model but it requires the word vectors to be uniformly distributed in the embedding space which generally is not true in practice [MT17]. Alternatively, [GAM17; AH19] adopts the paraphrase model that also does not fit practice. [EDH19], on the other hand, studies the geometry of embeddings that decomposes the shifted pointwise mutual information (PMI) matrix. Trager et al. [Tra+23] and Perera et al. [Per+23] decomposes embeddings into combinations of a smaller set of vectors that are more interpretable. On the other hand, similar to using vector orthogonality to represent (conditional) independence, kernel mean embeddings [Mua+17] are Hilbert space embeddings of distributions that can also be used to represent conditional independences [Son+09; SFG13]. It is a popular method for machine learning, and causal inference [Gre+05; Moo+09; GS20]. But unlike independence preserving embeddings, kernel mean embeddings use the kernel and do not explicitly construct finite-dimensional vector representations.

## 2 Independence Model and Markov Boundary

Let E be a finite set of embeddings with $|\,\mathrm{E}\,| = n$ and each embedding is of size $d$. Every embedding is a vector representation of a word. In other words, there exists a function $f$ that maps words to $n$ vectors in $\mathbb{R}^d$. As explained above, we might expect embeddings to encode "independence structures" between words. These independence structures are difficult to define formally, though the structure is similar to that of probabilistic conditional independence. We will use independence models as an abstract formalization of this structure.

### 2.1 Independence Model

Throughout this paper, we use many standard definitions and facts about graphical models and more generally, abstract independence models. A detailed overview of this material can be found, for instance, in [Lau96; Stu05].

Suppose $V$ is a finite set. In the case of embeddings, $V$ would be a set of vectors. An *independence model* $\perp\!\!\!\perp_\sigma$ is a ternary relation on $V$. Let $A, B, C, D$ be disjoint subsets of $V$. Then a *semi-graphoid* is an independence model that satisfies the following axioms:

(A1) (Symmetry) If $A \perp\!\!\!\perp_\sigma B|C$, then $B \perp\!\!\!\perp_\sigma A|C$;

(A2) (Decomposition) If $A \perp\!\!\!\perp_\sigma (B \cup D)|C$, then $A \perp\!\!\!\perp_\sigma B|C$ and $A \perp\!\!\!\perp_\sigma D|C$;

(A3) (Weak Union) If $A \perp\!\!\!\perp_\sigma (B \cup D)|C$, then $A \perp\!\!\!\perp_\sigma B|(C \cup D)$;

(A4) (Contraction) If $A \perp\!\!\!\perp_\sigma B|C$ and $A \perp\!\!\!\perp_\sigma D|(B \cup C)$, then $A \perp\!\!\!\perp_\sigma (B \cup D)|C$.

The independence model is a *graphoid* if it also satisfies

(A5) (Intersection) If $A \perp\!\!\!\perp_\sigma B|(C \cup D)$ and $A \perp\!\!\!\perp_\sigma C|(B \cup D)$, then $A \perp\!\!\!\perp_\sigma (B \cup C)|D$.

And, the graphoid is called a *compositional graphoid* if it also satisfies

(A6) (Composition) If $A \perp\!\!\!\perp_\sigma B|C$ and $A \perp\!\!\!\perp_\sigma D|C$, then $A \perp\!\!\!\perp_\sigma (B \cup D)|C$.

We also use $\mathcal{I}_\sigma(V)$ to be the set of conditional independent tuples under the independence model $\perp\!\!\!\perp_\sigma$. In other words, if $(A, B, C) \in \mathcal{I}_\sigma(V)$, then $A \perp\!\!\!\perp_\sigma B|C$ where $A, B, C$ are disjoint subsets of $V$.

**Probabilistic Conditional Independence ($\perp\!\!\!\perp_\mathrm{P}$)** Given a finite set of random variables $V$, probabilistic conditional independence over $V$ defines an independence model that satisfies (A1)-(A4) which means that probabilistic independence models are semi-graphoids. In general, however, they are not compositional graphoids. If the distribution has strictly positive density w.r.t. a product measure, then the intersection axiom is true. In this case, probabilistic independence models are graphoids. Still, in general, the composition axiom is not satisfied because pairwise independence does not imply

joint independence. One notable exception is when the distribution is regular multivariate Gaussian; then the probabilistic independence model is a compositional graphoid.

**Undirected Graph Separations ($\perp\!\!\!\perp_\mathrm{G}$)**   For a finite undirected graph $\mathcal{G} = (V, E)$. One can easily show that ordinary graph separation in undirected graphs is a compositional graphoid. The relations between probabilistic conditional independences and graph separations are well-studied in the graphical modeling literature [KF09; Lau96]. We recall a few important definitions here for completeness. Consider a natural bijection between graphical nodes and random variables. Then if $\mathcal{I}_G(V) \subseteq \mathcal{I}_P(V)$, we say the distribution $\mathcal{P}$ over $V$ satisfies the *Markov property* with respect to $\mathcal{G}$ and $\mathcal{G}$ is called an *I-map* of $\mathcal{P}$. An I-map $\mathcal{G}$ for $\mathcal{P}$ is minimal if no subgraph of $\mathcal{G}$ is also an I-map of $\mathcal{P}$. It is not difficult to show that there exists a minimal I-map $\mathcal{G}$ for any distribution $\mathcal{P}$.

*Remark* 1.  Not every compositional graphoid can be represented by an undirected graph. Sadeghi [Sad17] provides sufficient and necessary conditions for this.

**Partial Orthogonality ($\perp\!\!\!\perp_\mathrm{O}$)**   Let $V$ be a finite collection of vectors in $\mathbb{R}^d$. If $a \in V, b \in V$ and $C \subseteq V$, then we say that $a$ and $b$ are *partially orthogonal given $C$* if

$$a \perp\!\!\!\perp_\mathrm{O} b | C \iff \left\langle \mathrm{proj}_C^\perp[a], \ \mathrm{proj}_C^\perp[b] \right\rangle = 0,$$

where $\mathrm{proj}_C^\perp[a] = a - \mathrm{proj}_C[a]$ is the residual of $a$ after projection onto the span of $C$. It is not hard to verify that $\perp\!\!\!\perp_\mathrm{O}$ is a semi-graphoid that also satisfies the composition axiom (A6). When $V$ is a set of linearly independent vectors, then $\perp\!\!\!\perp_\mathrm{O}$ satisfies (A5) and thus is a compositional graphoid. Partial orthogonality has been studied under different names in the statistics literature for many decades. For example, if we replace Euclidean space with the $L^2$ space of random variables, partial orthogonality is equivalent to the well-known concept of *partial correlation* or *second-order independence* (Example 2.26 in [Lau20]). The concept of geometric orthogonality (Example 2.27 in [Lau20]) is closely related but does not always satisfy the intersection axiom. More recently, the concept of partial orthogonality in abstract Hilbert spaces was defined and studied extensively in [AAZ22]. Finally, when $V$ is a linearly independent collection of vectors, partial orthogonality yields a stronger independence model known as a *Gaussoid*, which is well-studied [e.g. LM07; BK19, and the references therein]. It is worth emphasizing that in the present setting of text embedding, we typically have $d \ll n$, and hence $V$ cannot be linearly independent.

## 2.2   Markov boundaries

Suppose $\perp\!\!\!\perp_\sigma$ is an independence model over a finite set $V$. Let $v_i$ be an element in $V$, then the Markov blanket $\mathcal{M}$ of $v_i$ is any subset of $V \setminus \{v_i\}$ such that

$$v_i \perp\!\!\!\perp_\sigma V \setminus (\{v_i\} \cup \mathcal{M}) | \mathcal{M}$$

A *Markov boundary* is a minimal Markov blanket.

A Markov boundary, by definition, always exists and can be an empty set. However, it might not be unique. It is well-known that *the intersection property is a sufficient condition to guarantee Markov boundaries are unique*. Thus, the Markov boundary is unique in any graphoid. The proof is presented here for completeness.

**Theorem 2.**  *If $\perp\!\!\!\perp_\sigma$ is a graphoid over $V$, then the Markov boundary is unique for any element in $V$.*

*Proof.* Let $v_i \in V$. Suppose $v_i$ has two distinct Markov boundaries $\mathcal{M}_1, \mathcal{M}_2$. Then they must be non-empty and $v_i \not\perp\!\!\!\perp_\sigma \mathcal{M}_1$, $v_i \not\perp\!\!\!\perp_\sigma \mathcal{M}_2$, $v_i \perp\!\!\!\perp_\sigma \mathcal{M}_2 | \mathcal{M}_1$, $v_i \perp\!\!\!\perp_\sigma \mathcal{M}_1 | \mathcal{M}_2$. By the intersection axiom, $v_i \perp\!\!\!\perp_\sigma \mathcal{M}_1 \cup \mathcal{M}_1$. Then by the decomposition axiom, $v_i \perp\!\!\!\perp_\sigma \mathcal{M}_1$ and $v_i \perp\!\!\!\perp_\sigma \mathcal{M}_2$ which is a contradiction.     □

*Remark* 3.  For any semi-graphoid, the intersection property is not a necessary condition for the uniqueness of Markov boundaries. See Remark 1 in [WW20].

The connection between orthogonal projection and graphoid axioms is well-known [Lau96; Daw01; Whi09]. But graphoid axioms find their primary applications in graphical models [Lau96]. In particular, there are many existing papers on Markov boundary discovery for graphical models [Tsa+03; Ali+10; SV16; GA21; TAS03; Pen+07]. They typically assume faithfulness or the

distributions are strictly positive, which are sufficient conditions for the intersection property and thus ensure unique Markov boundaries. As an important axiom for graphoids, the intersection property has also been thoroughly investigated [SMMR05; Pet15; Fin11]. But the intersection property rarely holds for embeddings (See Section 3), which means there could be multiple Markov boundaries. [SLA13; WW20] study this case for graphical models and causal inference.

# 3 Markov Boundary of Embeddings

As indicated in Section 2, partial orthogonality ($\perp\!\!\!\perp_O$) can be used as an independence model over vectors in Euclidean space and is a compositional semi-graphoid. Thus, one can use partial orthogonality to study embeddings, which are real vectors. When $n \leq d$ and the vectors in E are linearly independent, every vector in E has a unique Markov boundary by Theorem 2.

Unfortunately, when $d < n$, which happens in practice with embeddings as there are usually more objects to embed than the embedding dimension, there is a possibility of having multiple Markov boundaries. In fact, the main challenge with Markov boundary discovery for embeddings is that *the intersection property generally does not hold*, as opposed to graphical models where this property is commonly assumed [Tsa+03; Ali+10; SV16].

While the Markov boundary might not be unique, the following theorem says that all Markov boundaries of the target vector capture the same *"information"* about that vector.

**Theorem 4.** *Let partial orthogonality $\perp\!\!\!\perp_O$ be the independence model over a finite set of embedding vectors* E. *Suppose $\mathcal{M}_1, \mathcal{M}_2 \subseteq$ E are two distinct Markov boundaries of $v_i \in$ E, then,*

$$proj_{\mathcal{M}_1}[v_i] = proj_{\mathcal{M}_2}[v_i]$$

When $d \ll n$, then it is likely that the target embedding $v_i$ lies in the linear span of other embeddings (i.e, $v_i \in \text{span}(\text{E} \setminus \{v_i\})$), Corollary 5 below shows that, in this case, the span of any Markov boundary is precisely the subspace that contains $v_i$:

**Corollary 5.** *Let parital orthogonality $\perp\!\!\!\perp_O$ be the independence model over a finite set of embedding vectors* E. *Suppose $\mathcal{M}_1 \subseteq$ E is a Markov boundary of $v_i \in$ E and $v_i \in \text{span}(\text{E} \setminus \{v_i\})$, then,*

$$proj_{\mathcal{M}_1}[v_i] = v_i.$$

In other words, to find a Markov boundary of $v_i$, we need to find some vectors such that their linear combination is exactly $v_i$. This seems very strict but is necessary because the formal definition of the Markov boundary requires residual orthogonalities between $v_i$ and every other vector. In the sequel, we show how to relax the definition of the Markov boundary.

## 3.1 From Elementwise Orthogonality to Distributional Orthogonality

Corollary 5 suggests that the span of the Markov boundary for any target vector should contain that target vector. This is a consequence of the elementwise orthogonality constraint because the definition of the Markov boundary requires the residual of a target vector to be orthogonal to the residual of any test vector. The implicit assumption here is that embeddings are distortion-free and every non-zero correlation is meaningful. However, due to the inherent limitation of the embedding dimension—which often restricts the available space for storing all the orthogonal vectors—and noises introduced from training, embeddings are likely prone to distortion when compressed into a relatively small Euclidean space. In fact, we empirically show in Section 5.2 that inner products in embedding space do not necessarily respect semantic meanings faithfully. Therefore, the notion of elementwise orthogonality loses practical significance.

Instead of enforcing elementwise orthogonality, we relax the definition of the Markov boundary of embeddings such that intuitively, after projection, the residual of the target vector and the residuals of test vectors should be orthogonal in a distributional sense where the distribution is the empirical distribution over test vectors. To capture distributional orthogonalities, this paper focuses on the average of cosine similarities.

In particular, we have the following definition of *generalized Markov boundary* for partial orthogonality.

**Algorithm 1:** Approximate Algorithm to Find Generalized Markov Boundary

---

**Input:** $v, \mathrm{E}$
/* $\mathrm{E}$ is the set of all embeddings and $v \in \mathrm{E}$ is the target embedding    */
**Input:** $n_r, d_r, K$
/* $n_r$ is the number of sampled random subspaces, $d_r$ is the number of
   sampled vectors for each random subspace and $K$ is the number of
   candidate vectors to construct generalized Markov boundary    */
**Output:** $\mathcal{M} \subseteq \mathrm{E}$
/* $\mathcal{M}$ is the estimated generalized Markov boundary for $v$    */

**for** $i \leftarrow 1$ **to** $n_r$ **do**
    randomly sample a set of $d_r$ vectors $\mathcal{M}_i = \{v_k^i\}_{k=1}^{d_r} \subseteq (\mathrm{E} \setminus \{v\})$
    caculate $\mathrm{S^c}_{\mathcal{M}_i}(v, u)$ for all $u \in (\mathrm{E} \setminus \{v\})$
**end**
Find the top $K$ vectors $\{u_i\}_{i=1}^K$ with the highest $\sum_i \mathrm{S^c}_{\mathcal{M}_i}(v, u)$ .
Find the subset $\mathcal{M}$ of $\{u_i\}_{i=1}^K$ that has the lowest $\mathrm{S}_{\mathcal{M}}(v, \mathrm{E})$.
**Result:** $\mathcal{M}$

---

**Definition 6** (Generalized Markov Boundary for Partial Orthogonality)**.** Given a finite set $\mathrm{E}$ of embedding vectors. Let $v$ be an element in $\mathrm{E}$, then a *generalized Markov boundary* $\mathcal{M}$ of $v$ is a minimal subset of $\mathrm{E} \setminus \{v\}$ such that

$$\mathrm{S}_{\mathcal{M}}(v, \mathrm{E}) = \frac{1}{|\mathrm{E}_{\mathcal{M}}^v|} \sum_{u \in \mathrm{E}_{\mathcal{M}}^v} \mathrm{S^c}_{\mathcal{M}}(v, u) = 0$$

where $\mathrm{S^c}_{\mathcal{M}}(v, u)$ is the cosine similarity of $u$ and $v$ after projection and $\mathrm{E}_{\mathcal{M}}^v = \mathrm{E} \setminus (\{v\} \cup \mathcal{M})$. Specifically, $\mathrm{S^c}_{\mathcal{M}}(v, u) = \frac{\langle \mathrm{proj}_{\mathcal{M}}^{\perp}[v], \mathrm{proj}_{\mathcal{M}}^{\perp}[u] \rangle}{||\mathrm{proj}_{\mathcal{M}}^{\perp}[v]|| \cdot ||\mathrm{proj}_{\mathcal{M}}^{\perp}[u]||}$.

Intuitively, this suggests that, on average, there is no particular direction of residuals that have nontrivial correlations with the residual of the target embedding.

*Remark* 7. It is evident that the conventional definition of Markov boundary implies Definition 6 (Lemma 15 in Appendix A).

## 3.2   Finding Generalized Markov Boundary

With a formal definition of the generalized Markov boundary established, our objective is now to identify this boundary. One can always use brute force by enumerating all possible subsets of $\mathrm{E}$, but the algorithm would be infeasible when $|\mathrm{E}|$ is large.

Suppose $v \in \mathrm{E}$ is a target vector and $\mathcal{M}$ is its generalized Markov boundary, then we can write $v = v_{\perp} + v_{\parallel}$ where $v_{\perp} = \mathrm{proj}_{\mathcal{M}}^{\perp}[v]$ and $v_{\parallel} = \mathrm{proj}_{\mathcal{M}}[v]$. Intuitively, Definition 6 suggests that the residual of test vectors can appear in any direction relative to $v_{\perp}$. Therefore, if one samples random test vectors $\{u_i\}$, their span is likely to be close to $v_{\perp}$. In other words, the residual of $v$ after projection onto $\mathrm{span}(\{u_i\})$ should contain more information about the generalized Markov boundary direction $v_{\parallel}$.

This motivates the approximate method Algorithm 1. For any target embedding $v$, one first sample subspaces spanned by randomly selected embeddings. Embeddings that, on average have high cosine similarities with the target embedding after projecting onto orthogonal complements of previously sampled random subspaces, are considered to be candidates for the generalized Markov boundary. The final selection of generalized Markov boundary searches over these top $K$ candidates.

Empirically, for text embedding models like CLIP, random projections prove to be advantageous in revealing semantically related concepts. In Section 5.2, we provide several examples where, for a given target embedding, the embeddings that exhibit high correlation after random projections are more semantically meaningful compared to embeddings with merely high cosine similarity with the target embedding before projections.

# 4 Independence Preserving Embeddings (IPE)

In the previous sections, we discussed the Markov boundary of embeddings under the partial orthogonality independence model. In Section 5, we will test its effectiveness at capturing the "semantic independence structure" through experiments conducted on CLIP text embeddings. The belief is that the linear algebraic structure possesses the capacity to uphold the independence structure of semantic meanings.

A natural question to ask is: *Is it always possible to use vector space embeddings to preserve independence structures of interest?* In this section, we study the case for random variables. Consider an embedding function $f$ that maps a random variable $X$ to $f(X) \in \mathbb{R}^d$. Ideally, it is desirable for the partial orthogonalities of embeddings to mirror the conditional independences present in the joint distribution of $X$. We call such representations *independence preserving embeddings (IPE)* (Definition 8). In this section, we delve into the theoretical feasibility of these embeddings by initially demonstrating the construction of IPE and then showing how one can use random projection to reduce the dimension of IPE. We believe that studying IPE lays the theoretical foundation to understand embedding models in general.

**Definition 8** (Independence Preserving Embedding Map). Let $V$ be a finite set of random variables with distribution $P$. A function $f : V \to \mathbb{R}^d$ is called an *independence preserving embedding map* (IPE map) if
$$\mathcal{I}_O(f(V)) \subseteq \mathcal{I}_P(V).$$
An IPE map is called a *faithful* IPE map if
$$\mathcal{I}_O(f(V)) = \mathcal{I}_P(V).$$

## 4.1 Existence and Universality of IPE Maps

We first show that for *any distribution $P$* over random variables $V$, we can construct an IPE map.

For any distribution $P$ over $V$, there exists a minimal $I$-map $\mathcal{G} = (V, E)$ such that $\mathcal{I}_G(V) \subseteq \mathcal{I}_P(V)$ (See Section 2). We will use $\mathcal{G}_P$ to be a minimal $I$-map of $P$ and $\mathrm{adj}(\mathcal{G}_P)$ to be the adjacency matrix of $\mathcal{G}_P$. We further define $\mathrm{adj}_\varepsilon(\mathcal{G}_P)$ to be an *adjusted adjacency matrix* with $\varepsilon \in \mathbb{R}$ where

$$\mathrm{adj}_\varepsilon(\mathcal{G}_P) = \mathbb{1} + \varepsilon \, \mathrm{adj}(\mathcal{G}_P)$$

and $\mathbb{1}$ is the identity matrix.

Ideally, this matrix is invertible, however, it turns out that not every $\varepsilon$ produces an invertible $\mathrm{adj}_\varepsilon(\mathcal{G}_P)$. We therefore define the following *perfect perturbation factor*. For any matrix $A \in \mathbb{R}^{n \times n}$, define $A_{\mathcal{I}, \mathcal{J}}$ to be the submatrix of $A$ with row and column indices from $\mathcal{I}$ and $\mathcal{J}$, respectively. If $\mathcal{I} = \mathcal{J}$, the submatrix is called a *principal submatrix* and we denote it simply as $A_\mathcal{I}$.

**Definition 9** (Perfect Perturbation Factor). For a given graph $\mathcal{G} = (V, E)$ where $n = |V|$, $\varepsilon$ is called a *perfect perturbation factor* if (1) $\mathrm{adj}_\varepsilon(\mathcal{G}_P)$ is invertible and (2) for any $\mathcal{I} \subseteq [n]$, $(\mathrm{adj}_\varepsilon(\mathcal{G}_P)_\mathcal{I})^{-1}_{ij} = 0$ if and only if $v_{\mathcal{I}_i} \perp\!\!\!\perp_G v_{\mathcal{I}_j} | \{v_k : k \notin \mathcal{I}\}$ where $\mathcal{I}_i$ is the $i$th element of $\mathcal{I}$.

**Theorem 10.** *Let $V$ be a finite set of random variables with distribution $P$. $\mathcal{G}_P$ is a minimal $I$-map of $P$. Let $A$ be equal to $\mathrm{adj}_\varepsilon(\mathcal{G}_P)^{-1}$ with eigen decomposition $A = U\Sigma U^T$. If $\varepsilon$ is a perfect perturbation factor, then the function $f$ with*
$$f(v_i) = U_i \Sigma^{1/2}$$
*is an IPE map of $P$ where $v_i$ is a random variable in $V$ and $U_i$ is the $i$-th row of $U$. Furthermore, if $P$ is faithful to $\mathcal{G}_P$, then $f$ is a faithful IPE map for $\mathcal{P}$.*

*Remark* 11. One can always normalize these embeddings to have unit norms without changing the partial orthogonality structures.

Finding a perfect perturbation factor might seem daunting, but the following lemma, which is a direct consequence of Theorem 1 in Lněnička and Matúš [LM07], shows that almost every $\varepsilon$ is a perfect perturbation factor.

**Lemma 12.** *For any simple graph $G$, $\varepsilon$ is perfect for all but finitely many $\varepsilon \in \mathbb{R}$.*

## 4.2 Dimension Reduction of IPE

Theorem 10 shows how to learn a perfect IPE but it requires the dimension of embeddings to be the same as the number of variables in $V$. In the worst case, this is inevitable for a faithful IPE map: If the random variables in $V$ are mutually independent, then we need at least $|V|$ dimensions in the embedding space to contain $V$ orthogonal vectors.

But this is not practical. Suppose we want to embed millions of random variables (e.g. tokens) in a vector space, having the dimension of each embedding be in the magnitude of millions is less than ideal. Therefore, one needs to do dimension reduction.

In this section, we show that by using random projection, the partial orthogonalities induced by Markov boundaries are preserved approximately. Intuitively, this is guaranteed by the Johnson-Lindenstrauss lemma [Vem05].

**Theorem 13.** *Let $U$ be a set of vectors in $\mathbb{R}^n$ where $n = |U|$ and every vector is a unit vector. Let $\Sigma$ be a matrix in $\mathbb{R}^{n \times n}$ where $\Sigma_{ij} = \langle u_i, u_j \rangle$. Assume $\lambda_1 = \lambda_{\min}(\Sigma) > 0$. Then there exists a mapping $g : \mathbb{R}^n \to \mathbb{R}^k$ where $k = \lceil 20 \log(2n)/(\varepsilon')^2 \rceil$ with $\varepsilon' = \min\{1/2, \varepsilon/C, \lambda_1/2r^2\}$ and $\varepsilon \in (0,1)$ such that for any $u_i \in U$ with its unique Markov boundary $M_i \subseteq U$ and any $u_j \in U \setminus (\{u_i\} \cup M_i)$, we have*

$$\left| \left\langle \text{proj}^{\perp}_{g(M_i)}[g(u_i)], \text{proj}^{\perp}_{g(M_i)}[g(u_j)] \right\rangle \right| \leq \varepsilon$$

*where $r_i = |M_i|$, $r = \max_i |M_i|$ and $C = (r+1)^3 \left( \frac{2\lambda_{\max}(\Sigma) + 2(r+1)^2}{\lambda_{\min}(\Sigma)} \right)^r$.*

Theorem 13 shows that as long as the partial orthogonality structure of embeddings is sparse in the sense that the size of the Markov boundary for each embedding is small. Then one can reduce the dimension of the embedding and the residuals of target and test vectors after projection onto the Markov boundary are *almost orthogonal*.

*Remark* 14. The assumption in Theorem 13 is satisfied by the construction of IPE in Section 4.1.

## 5 Experiments

One of the central hypotheses of the paper is that the partial orthogonality of embeddings, and its byproduct generalized Markov boundary, carry semantic information. To verify this claim, we provide both *quantitative* and *qualitative* experiments. Throughout this section, we consider the set of normalized embeddings E that represent the 49815 words in the Brown corpus [FK79]. For each target embedding of a word, under any experiment setting, we automatically filter words, whose embeddings have 0.9 or above cosine similarities with the target embedding, or words, whose Wu-Palmer similarity measure with the target word is almost 1. The purpose of this filtering step is to prevent the inclusion of synonyms.

### 5.1 Semantic Structure of Partial Orthogonality

To examine the rule of partial orthogonality, nine categories are chosen, each with 10 words in it (See Table 3 in Appendix B). Specifically, each word within a given category is a hyponym for that category in WordNet [Mil95]. We assess how much, on average, the cosine similarities between words within each category decrease when conditioned on these different nine categories. By conditioning, we use the clip embedding of the category word of interest and project out the subspace of that clip embedding. The results are shown in Figure 2. We normalize reduction values by sampling 10,000 embeddings and calculating the mean and standard deviation of cosine reductions between these embeddings. It is apparent that on average, cosine similarities of intra-category words decrease more than inter-category words. One interesting finding is that when conditioned on the category word "food", the average similarities between word pairs in "beverage" also drop considerably. We suspect this is because one synset of "food" is also a hypernom of "beverage". Although words in the "food" category are chosen to mean solid food, it could also mean nutrient which also encompasses the meaning of "beverage".

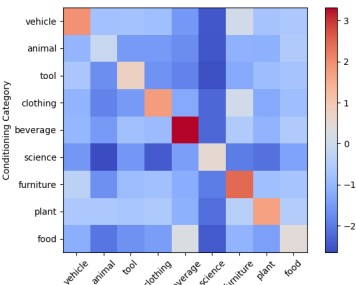

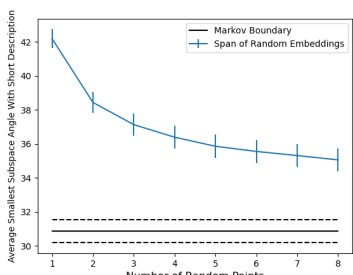

**Figure 2:** Experiments show conditioning on the category word, cosine similarities of intra-category words decrease more than inter-category words. Each row shows the (normalized) average cosine similarities reduction between words within each category when conditioned on the category word of that row.

**Figure 3:** Experiments show that learned generalized Markov boundaries have on average smaller principal angles with the description embedding compared to subspaces spanned by randomly selected embeddings. The standard errors are over 50 examples.

## 5.2 Sampling Random Subspaces

The first step of Algorithm 1 is to find embeddings that have high similarities with the target embeddings even after projecting onto orthogonal complements of subspaces spanned by randomly selected embeddings. It turns out that this step can reveal semantic meanings. In this section, we design experiments to show both quantitatively and qualitatively that embeddings of words that remain highly correlated with the target embedding after projection are semantically closer to the target word. In various experimental configurations, we employ 10 sets of 50 randomly chosen embeddings to form random projection subspaces for each target embedding. Qualitatively, Table 4 in Appendix B gives a few examples showing that the words that on average remain highly correlated with the target word tend to possess greater semantic significance. Quantitatively, we calculate the average Wu-Palmer similarities between target words and the top 10 correlated words before and after random projections. We conduct experiments on 1000 random words as well as 300 common nouns provided by ChatGPT. The results are shown in Table 4 verify our claims. This set of experiments also indirectly shows that the embeddings are noisy and that generalized Markov boundaries are indeed needed.

**Table 1:** Experiments show that the top 10 words that have, on average, high correlations with target words after projecting onto the orthogonal complements of randomly selected linear subspaces have higher Wu-Palmer similarities with the target words than the top 10 highly correlated words without projections. This table contains average Wu-Palmer similarities with standard errors.

| Target | Before Projection | After Projection |
|---|---|---|
| Random Words | $0.223 \pm 0.006$ | $0.245 \pm 0.007$ |
| Common Nouns | $0.343 \pm 0.008$ | $0.422 \pm 0.008$ |

## 5.3 Generalized Markov Boundaries

We first demonstrate that Algorithm 1 can find generalized Markov boundaries. The experiments are run over 1000 randomly selected words. In particular, Table 2 shows that with a relatively small candidate set, the algorithm can already approximate generalized Markov boundaries well, suggesting that the size of generalized Markov boundaries for CLIP text embeddings should be small.

**Semantic Meanings of Markov Boundaries** The estimated generalized Markov boundaries returned by Algorithm 1 is a set of embeddings. It is reasonable to anticipate that the linear spans of these embeddings hold semantic meanings. To evaluate this hypothesis, we propose to calculate the smallest principal angles [KA02] between the span of generalized Markov boundaries and the span of selected embeddings that are meaningful to the target word.

We again conducted both quantitative and qualitative experiments. Qualitatively, Figure 4 in Appendix B give a few examples comparing target words' generalized Markov boundaries with the span of selected embeddings. For instance, the generalized Markov boundary of 'car' is more aligned with the subspace spanned by embeddings of 'road' and 'vehicle' than the span of 'sea' and 'boat' and randomly selected subspaces. This suggests that the estimated generalized Markov boundaries hold semantic significance. To verify this quantitatively, we ask ChatGPT to provide a list of common nouns with short descriptions (selected examples are provided in Table 5). We then use CLIP text embedding to convert the description sentence into one vector and compare the smallest angle between the description vector with generalized Markov boundaries and random linear spans. Figure 3 shows that the generalized Markov boundaries are more semantically meaningful than random subspaces.

**Table 2:** With relatively small number of $K$, the average $S_{\mathcal{M}}(v, E)$ is small. The standard errors are over 1000 experiments.

| $K$ | 1 | 3 | 5 | 8 | 10 |
|---|---|---|---|---|---|
| Average $S_{\mathcal{M}}(v, E)$ | $0.345\pm0.03$ | $0.128\pm0.03$ | $0.054\pm0.002$ | $0.015\pm0.002$ | $0.008\pm0.001$ |

## 6  Conclusion

This paper studies the role of partial orthogonality in analyzing embeddings. Specifically, we extend the idea of Markov boundaries to embedding space. Unlike Markov boundaries in graphical models, the boundaries for embeddings are not guaranteed to be unique. We propose alternative relaxed definitions of Markov boundaries for practical use. Empirically, these tools prove to be useful in finding the semantic meanings of embeddings. We also introduce the concept of independence preserving embeddings where embeddings use partial orthogonalities to preserve the conditional independence structures of random variables. This opens the door for substantial future work. In particular, one promising theoretical direction is to study if CLIP text embeddings preserve the structures in the training distributions.

## 7  Acknowledgments

This work is supported by ONR grant N00014-23-1-2591, Open Philanthropy, NSF IIS-1956330, NIH R01GM140467, and the Robert H. Topel Faculty Research Fund at the University of Chicago Booth School of Business.

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

# A  Additional Proofs

**Lemma 15.** *Let $v$ be an element in* E*, and $\mathcal{M}$ a Markov boundary of $v$, then $\mathcal{M}$ is also a generalized Markov boundary.*

*Proof.* By definition, for any $u \in V \setminus (\{v\} \cup \mathcal{M})$, $\mathrm{S^c}_{\mathcal{M}}(v, u) = 0$. Therefore, $\mathcal{M}$ is also a generalized Markov boundary. $\square$

**Theorem 4.** *Let partial orthogonality $\perp\!\!\!\perp_\mathrm{O}$ be the independence model over a finite set of embedding vectors* E*. Suppose $\mathcal{M}_1, \mathcal{M}_2 \subseteq$ E are two distinct Markov boundaries of $v_i \in$ E, then,*

$$proj_{\mathcal{M}_1}[v_i] = proj_{\mathcal{M}_2}[v_i]$$

*Proof.* To slightly abuse notation, we also use $\mathcal{M}_1$ to be a matrix where each column is an element in $\mathcal{M}_1$. We define $\mathcal{M}_2$ similarly.

Because $\mathcal{M}_1$ and $\mathcal{M}_2$ are two distinct Markov boundaries, they must not be empty. Therefore, $v_i \not\perp\!\!\!\perp_\mathrm{O} \mathcal{M}_1$ and $v_i \not\perp\!\!\!\perp_\mathrm{O} \mathcal{M}_2$. By the definition of Markov boundary, we also have $v_i \perp\!\!\!\perp_\mathrm{O} \mathcal{M}_1 \mid \mathcal{M}_2$ and $v_i \perp\!\!\!\perp_\mathrm{O} \mathcal{M}_2 \mid \mathcal{M}_1$. Note that $\mathcal{M}_2$ and $\mathcal{M}_1$ must have full rank, otherwise, they are not minimal.

Thus,

$$\langle \mathrm{proj}^{\perp}_{\mathcal{M}_1}[v_i],\ \mathrm{proj}^{\perp}_{\mathcal{M}_1}[v_j] \rangle = 0,\ \forall v_j \in \mathcal{M}_2$$
$$\iff \langle \mathrm{proj}^{\perp}_{\mathcal{M}_1}[v_i],\ v_j \rangle = 0,\ \forall v_j \in M_2$$
$$\iff v_i^T \mathcal{M}_1 (\mathcal{M}_1^T \mathcal{M}_1)^{-1} \mathcal{M}_1^T v_j = v_i^T v_j\ \forall v_j \in M_2$$
$$\iff v_i^T \mathcal{M}_1 (\mathcal{M}_1^T \mathcal{M}_1)^{-1} \mathcal{M}_1^T \mathcal{M}_2 = v_i^T \mathcal{M}_2$$

With (compact) singular value decomposition, we have $\mathcal{M}_1 = U_1 \Sigma_1 V_1^T$ and $\mathcal{M}_2 = U_2 \Sigma_2 V_2^T$. Then,

$$v_i^T \mathcal{M}_1 (\mathcal{M}_1^T \mathcal{M}_1)^{-1} \mathcal{M}_1^T \mathcal{M}_2 = v_i^T U_1 U_1^T \mathcal{M}_2 = v_i^T \mathcal{M}_2$$
$$\iff v_i^T U_1 U_1^T U_2 = v_i^T U_2$$

Similarly,

$$v_i^T U_2 U_2^T U_1 = v_i^T U_1$$

Therefore,

$$v_i^T U_1 U_1^T U_2 U_2^T = v_i^T U_2 U_2^T$$

In other words,

$$proj_{\mathcal{M}_2}[proj_{\mathcal{M}_1}[v_i]] = proj_{\mathcal{M}_2}[v_i]$$

On the other hand, $v_i \not\perp\!\!\!\perp_\mathrm{O} \mathcal{M}_1$.

$$proj_{\mathcal{M}_1}[v_i] = U_1 U_1^T v_i \neq 0$$

Similarly,

$$proj_{\mathcal{M}_2}[v_i] = U_2 U_2^T v_i \neq 0$$

Therefore, we must have,

$$proj_{\mathcal{M}_1}[v_i] \in \mathrm{span}(\mathcal{M}_2)$$

which means,

$$proj_{\mathcal{M}_2}[proj_{\mathcal{M}_1}[v_i]] = proj_{\mathcal{M}_1}[v_i] = proj_{\mathcal{M}_2}[v_i]$$

$\square$

**Corollary 5.** *Let parital orthogonality $\perp\!\!\!\perp_\mathrm{O}$ be the independence model over a finite set of embedding vectors* E*. Suppose $\mathcal{M}_1 \subseteq$ E is a Markov boundary of $v_i \in$ E and $v_i \in \mathrm{span}(\mathrm{E} \setminus \{v_i\})$, then,*

$$proj_{\mathcal{M}_1}[v_i] = v_i.$$

*Proof.* Because $v_i \in \text{span}(\text{E} \setminus \{v_i\})$, then $v_i = \sum_{k=1}^{m} \alpha_k v_k$ with $\text{E}' = \{v_1, ..., v_m\} \subseteq \text{E}$. Since $\mathcal{M}_1$ is a Markov boundary of $v_i$,

$$v_i^T \mathcal{M}_1 (\mathcal{M}_1^T \mathcal{M}_1)^{-1} \mathcal{M}_1^T v_k = v_i^T v_k \; \forall v_k \in \text{E}'$$

$$v_i^T \mathcal{M}_1 (\mathcal{M}_1^T \mathcal{M}_1)^{-1} \mathcal{M}_1^T \sum_{k=1}^{m} \alpha_k v_k = v_i^T \sum_{k=1}^{m} \alpha_k v_k$$

$$\langle proj_{\mathcal{M}_1}[v_i], v_i \rangle = \langle v_i, v_i \rangle$$

Thus, $proj_{\mathcal{M}_1}[v_i] = v_i$.

$\square$

## A.1 Construction of IPE Map

**Theorem 10.** *Let $V$ be a finite set of random variables with distribution $P$. $\mathcal{G}_P$ is a minimal I-map of $P$. Let $A$ be equal to $\text{adj}_\varepsilon(\mathcal{G}_P)^{-1}$ with eigen decomposition $A = U\Sigma U^T$. If $\varepsilon$ is a perfect perturbation factor, then the function $f$ with*

$$f(v_i) = U_i \Sigma^{1/2}$$

*is an IPE map of $P$ where $v_i$ is a random variable in $V$ and $U_i$ is the $i$-th row of $U$. Furthermore, if $P$ is faithful to $\mathcal{G}_P$, then $f$ is a faithful IPE map for $\mathcal{P}$.*

*Proof.* Let $|V| = n$ and, to slightly abuse notation, we use index $i \in [n]$ to mean the vertex $v_i$ and the $i$-th embedding. And we use $A_{V_1,V_2}$ where $V_1, V_2 \subseteq V$ to mean the submatrix of $A_{\{i:v_i \in V_1\},\{i:v_i \in V_2\}}$.

Because $G_p$ is a miminal $I$-map of $P$, we have $\mathcal{I}_G(V) \subseteq \mathcal{I}_P(V)$.

We just need to show $\mathcal{I}_G(V) = \mathcal{I}_O(f(V))$. And if $P$ is faithful to $G_p$, then $\mathcal{I}_O(f(V)) = \mathcal{I}_G(V) = \mathcal{I}_P(V)$.

If $v_i \perp\!\!\!\perp_\text{G} v_j | V'$ where $V' \subseteq V$, let $V^c = V \setminus V'$, then

$$A = \begin{pmatrix} A_{V^c}, A_{V^c,V'} \\ A_{V',V^c}, A_{V'} \end{pmatrix}$$

On the other hand, if $f(v_i) \perp\!\!\!\perp_\text{O} f(v_j) | f(V')$, then

$$f(v_i)^T f(v_j) - f(v_i)^T f(V')(f(V')^T f(V'))^{-1} f(V')^T f(v_j) = 0 \tag{A.1}$$

Note that by our construction, $(f(V')^T f(V'))^{-1} = A_{V'}$ is invertible. We can write (A.1) as follows:

$$f(v_i)^T f(v_j) - f(v_i)^T f(V')(f(V')^T f(V'))^{-1} f(V')^T f(v_j) = A_{i,j} - A_{i,V'} A_{V'}^{-1} A_{V',j}$$

By Schur's complement, we have that

$$(\text{adj}_\varepsilon(G_P)_{V^c})^{-1} = (A^{-1})_{V^c}^{-1} = A_{V^c} - A_{V^c,V'} A_{V'}^{-1} A_{V',V^c}$$

Becasue $\varepsilon$ is a perfect perturbation factor, by definition, $f(v_i) \perp\!\!\!\perp_\text{O} f(v_j) | f(V')$ if and only if $v_i \perp\!\!\!\perp_\text{G} v_j | V'$. By the compositional property, we have that $\mathcal{I}_G(V) = \mathcal{I}_O(f(V))$. $\square$

**Lemma 12.** *For any simple graph $G$, $\varepsilon$ is perfect for all but finitely many $\varepsilon \in \mathbb{R}$.*

*Proof.* This is a direct consequence of Theorem 1 in [LM07]. $\square$

## A.2   Dimension Reduction of IPE

**Theorem 13.** *Let $U$ be a set of vectors in $\mathbb{R}^n$ where $n = |U|$ and every vector is a unit vector. Let $\Sigma$ be a matrix in $\mathbb{R}^{n \times n}$ where $\Sigma_{ij} = \langle u_i, u_j \rangle$. Assume $\lambda_1 = \lambda_{\min}(\Sigma) > 0$. Then there exists a mapping $g : \mathbb{R}^n \to \mathbb{R}^k$ where $k = \lceil 20 \log(2n)/(\varepsilon')^2 \rceil$ with $\varepsilon' = \min\{1/2, \varepsilon/C, \lambda_1/2r^2\}$ and $\varepsilon \in (0, 1)$ such that for any $u_i \in U$ with its unique Markov boundary $M_i \subseteq U$ and any $u_j \in U \setminus (\{u_i\} \cup M_i)$, we have*

$$\left| \left\langle \mathrm{proj}^{\perp}_{g(M_i)}[g(u_i)], \mathrm{proj}^{\perp}_{g(M_i)}[g(u_j)] \right\rangle \right| \leq \varepsilon$$

*where $r_i = |M_i|$, $r = \max_i |M_i|$ and $C = (r+1)^3 \left( \frac{2\lambda_{\max}(\Sigma) + 2(r+1)^2}{\lambda_{\min}(\Sigma)} \right)^r$.*

*Proof.* Let $g$ be linear map of Lemma 16 with error parameter $\varepsilon' \in (0, \frac{1}{2})$. For convenience, let $\tilde{u}_i = g(u_i)$, $\tilde{u}_j = g(u_j)$ and $\tilde{M}_i = g(M_i)$. Let $r_i = |M_i|$. To slightly abuse notation, we use $M_i$ and $\tilde{M}_i$ to also mean matrices where each column is an element in the set. Furthermore, we also define $\tilde{\Sigma}$ to be $\tilde{\Sigma}_{ij} = \langle \tilde{u}_i, \tilde{u}_j \rangle$. We use $\Sigma_{A,B}$ where $A, B \subseteq U$ to be a submatrix where the row indices are from $A$ and the column indices are from $B$, and when $A = B$, we just use $\Sigma_A$ for simplicity. In particular, let $\Sigma_{(i,M_i),(j,M_i)} = \begin{pmatrix} \Sigma_{i,j}, & \Sigma_{i,M_i} \\ \Sigma_{M_i,j}, & \Sigma_{M_i} \end{pmatrix}$. And we can define a similar thing for $\tilde{\Sigma}$.

We first want to find $\varepsilon'$ such that $\tilde{\Sigma}_{M_i}$ is non-singular for all $i \in |U|$. Note that for any $u_i \in U$, we know that by Weyl's inequality for eigenvalues [HJ12],

$$|\lambda_{\min}(\tilde{\Sigma}_{M_i}) - \lambda_{\min}(\Sigma_{M_i})| \leq ||\tilde{\Sigma}_{M_i} - \Sigma_{M_i}|| \leq ||\tilde{\Sigma}_{M_i} - \Sigma_{M_i}||_F \leq r^2 \varepsilon'$$

Thus,

$$\lambda_{\min}(\tilde{\Sigma}_{M_i}) \geq \lambda_{\min}(\Sigma_{M_i}) - r^2 \varepsilon \geq \lambda_{\min}(\Sigma) - r^2 \varepsilon = \lambda_1 - r^2 \varepsilon'$$

Therefore, if we want $\lambda_{\min}(\tilde{\Sigma}_{M_i}) > \frac{\lambda_1}{2}$ we need $\varepsilon' < \frac{\lambda_1}{2r^2}$.

On the other hand, because $M_i$ is an Markov boundary for $u_i$, we have

$$u_i^T u_j - u_i^T M_i (M_i^T M_i)^{-1} M_i^T u_j = 0 \tag{A.2}$$

Note that $M_i$ must be full rank. Otherwise, we can find a subset of $M_i$ to be the Markov boundary. And there is a different way to write this. Remember that $\Sigma_{(i,M_i),(j,M_i)} = \begin{pmatrix} \Sigma_{i,j}, & \Sigma_{i,M_i} \\ \Sigma_{M_i,j}, & \Sigma_{M_i} \end{pmatrix}$. Using Schur's complement, we have that

$$\det(\Sigma_{(i,M_i),(j,M_i)}) = \det(\Sigma_{M_i}) \det(\Sigma_{i,j} - \Sigma_{i,M_i}^T (\Sigma_{M_i})^{-1} \Sigma_{j,M_i})$$
$$= \det(\Sigma_{M_i})(u_i^T u_j - u_i^T M_i (M_i^T M_i)^{-1} M_i^T u_j)$$

We want to estimate the following:

$$|\langle \mathrm{proj}^{\perp}_{g(M_i)}[g(u_i)], \mathrm{proj}^{\perp}_{g(M_i)}[g(u_j)] \rangle| = |\tilde{u}_i^T \tilde{u}_j - \tilde{u}_i^T \tilde{M}_i (\tilde{M}_i^T \tilde{M}_i)^{-1} \tilde{M}_i^T \tilde{u}_j|$$
$$= \left| \frac{\det(\tilde{\Sigma}_{(i,M_i),(j,M_i)})}{\det(\tilde{\Sigma}_{M_i})} \right| \tag{A.3}$$

We already know that $\det(\tilde{\Sigma}_{M_i}) > (\frac{\lambda_1}{2})^{r_i}$. On the other hand, by Theorem 2.12 in [IR08], we have that

$$|\det(\tilde{\Sigma}_{(i,M_i),(j,M_i)})| = |\det(\tilde{\Sigma}_{(i,M_i),(j,M_i)}) - \det(\Sigma_{(i,M_i),(j,M_i)})|$$
$$\leq (r_i + 1)||\tilde{\Sigma}_{(i,M_i),(j,M_i)} - \Sigma_{(i,M_i),(j,M_i)}|| \max\{||\Sigma_{(i,M_i),(j,M_i)}||, ||\tilde{\Sigma}_{(i,M_i),(j,M_i)}||\}^{r_i}$$

By Weyl's inequality for singular values, we have that

$$||\tilde{\Sigma}_{(i,M_i),(j,M_i)}|| = \sigma_{\max}(\tilde{\Sigma}_{(i,M_i),(j,M_i)}) \leq \sigma_{\max}(\Sigma_{(i,M_i),(j,M_i)}) + (r_i + 1)^2 \varepsilon'$$
$$\leq \lambda_{\max}(\Sigma) + (r_i + 1)^2 \varepsilon' \leq \lambda_{\max}(\Sigma) + (r_i + 1)^2$$

Thus,
$$|\det(\tilde{\Sigma}_{(i,M_i),(j,M_i)})| \le (r_i+1)(r_i+1)^2\varepsilon'(\lambda_{\max}(\Sigma)+(r_i+1)^2)^{r_i}$$

And,
$$|\frac{\det(\tilde{\Sigma}_{(i,M_i),(j,M_i)})}{\det(\tilde{\Sigma}_{M_i})}| \le \varepsilon'\frac{(r_i+1)^3(\lambda_{\max}(\Sigma)+(r_i+1)^2)^{r_i}}{(\frac{\lambda_1}{2})^{r_i}}$$

Let $C = (r+1)^3(\frac{2\lambda_{\max}(\Sigma)+2(r+1)^2}{\lambda_{\min}(\Sigma)})^r$. Then,
$$|\frac{\det(\tilde{\Sigma}_{(i,M_i),(j,M_i)})}{\det(\tilde{\Sigma}_{M_i})}| \le \varepsilon'C$$

Let $\varepsilon' = \min\{\frac{1}{2}, \frac{\varepsilon}{C}, \frac{\lambda_1}{2r^2}\}$ and $k = \lceil\frac{20\log(2n)}{(\varepsilon')^2}\rceil$, we have that
$$|\langle \mathrm{proj}^{\perp}_{g(M_i)}[g(u_i)], \mathrm{proj}^{\perp}_{g(M_i)}[g(u_j)]\rangle| \le \varepsilon$$

$\square$

**Lemma 16.** *Let $\varepsilon \in (0, \frac{1}{2})$. Let $V \subseteq \mathbb{R}^d$ be a set of $n$ points and $k = \lceil\frac{20\log(2n)}{\varepsilon^2}\rceil$, there exists a linear mapping $g : \mathbb{R}^d \to \mathbb{R}^k$ such that for all $u, v \in V$:*
$$|\langle g(u), g(v)\rangle - \langle u, v\rangle| \le \varepsilon$$

*Proof.* The proof is an easy extension of the JL lemma [Vem05] by adding all the $-v_i$ for all $v_i \in V$ into the set $V$. $\square$

# B   Additional Experiments, Figures and Tables

**Table 3:** 9 categories of words used to test the semantic meaning of partial orthogonality

| Category | Words in Category |
|---|---|
| 'vehicle' | 'car', 'bicycle', 'skateboard', 'motorcycle', 'helicopter', 'truck', 'boat', 'airplane', 'submarine', 'scooter' |
| 'animal' | 'lion', 'dolphin', 'eagle', 'dog', 'elephant', 'cat', 'rat', 'giraffe', 'bird', 'tiger' |
| 'tool' | 'hammer', 'screwdriver', 'wrench', 'pliers', 'hacksaw', 'drill', 'chisel', 'plunger', 'trowel', 'cutter' |
| 'clothing' | 'shirt', 'pants', 'dress', 'sweater', 'jacket', 'hat', 'socks', 'gloves', 'scarf', 'vest' |
| 'beverage' | 'coffee', 'tea', 'soda', 'lemonade', 'milk', 'wine', 'beer', 'sake', 'smoothie', 'nectar' |
| 'science' | 'biology', 'ecology', 'genetics', 'chemistry', 'physics', 'geology', 'mathematics', 'linguistics', 'psychology', 'cryptography' |
| 'furniture' | 'couch', 'bed', 'cabinet', 'dresser', 'hallstand', 'lamp', 'bench', 'chair', 'table', 'closet' |
| 'plant' | 'daisy', 'pine', 'iris', 'lily', 'oak', 'tulip', 'fern', 'rose', 'bamboo', 'cactus' |
| 'food' | 'chocolate', 'meat', 'steak', 'pasta', 'fish', 'brisket', 'sausage', 'loaf', 'roe', 'lobster' |

**Table 4:** Experiments show that top correlated words with target words after projecting onto the orthogonal complements of randomly selected linear subspaces are more semantically meaningful

| Target | Top Correlated Words Before Projection | Top Correlated Words After Projection |
|---|---|---|
| 'eggplant' | 'potato', 'banana', 'grape', 'vegetable' 'bananas', 'tomato', 'espagnol' 'eternal', 'potatoes', 'e.g.' | 'grape', 'purple-black', 'purple', 'turnips' 'plum', 'lilac', 'vegetable' 'vegetables' 'banana', 'ultra-violet' |
| 'king' | 'mister', 'bossman', 'thet', 'thatt' 'beast', 'killed', 'yesiree' 'bossed', 'outdo', 'queen's' | 'royalty', 'sport-king', 'bossman', 'kingan' 'mister', 'prince's', 'princess' 'princes' 'handsomest', 'ruling' |
| 'advise' | 'spoken', 'askin', 'concur', 'applies' 'said', 'according', 'astute' 'pertinent', 'evident', 'preached' | 'guidelines', 'guidance', 'tips', 'motto' 'motivating', 'encourages', 'advising' 'advisory' 'self-help', 'reminder' |
| 'work-out' | 'healthy', 'weights', 'worked', 'time-on-the-job' 'on-the-job', 'work-success', 'busy-work' 'out'n', 'healthiest', 'hardworking' | 'gym', 'weights', 'footing', 'running' 'jogs', 'dumbbells', 'conditioning' 'body-building' 'runing', 'pumped-up' |
| 'poem' | '!', 'ya', 'eh', 'yes', ';' 'mem', 'oh', ')', 'poignant', 'hee' | 'poems', 'poetizing', 'poetry's', 'rhyming' 'sonnet', 'lyrics', 'recited' 'poetically' 'sonnets', 'rhyme' |

**Table 5:** Selected examples provided by ChatGPT when asked "give me a list of 50 common nouns, each with a short description, and the first one is eggplant"

| Target Word | Description Sentence |
| --- | --- |
| 'eggplant' | 'A purple or dark-colored vegetable with a smooth skin, often used in cooking and known for its mild flavor.' |
| 'dog' | 'A domesticated mammal often kept as a pet or used for various purposes.' |
| 'book' | 'A physical or digital publication containing written or printed content.' |
| 'car' | 'A motorized vehicle used for transportation on roads.' |
| 'tree' | 'A woody perennial plant with a main trunk and branches, usually producing leaves.' |
| 'house' | 'A building where people live, providing shelter and accommodation.' |
| 'computer' | 'An electronic device used for processing and storing data, and performing various tasks.' |
| 'cat' | 'A small domesticated carnivorous mammal commonly kept as a pet.' |
| 'chair' | 'A piece of furniture designed for sitting on, often with a backrest and four legs.' |
| 'phone' | 'A communication device that allows voice calls and text messaging.' |

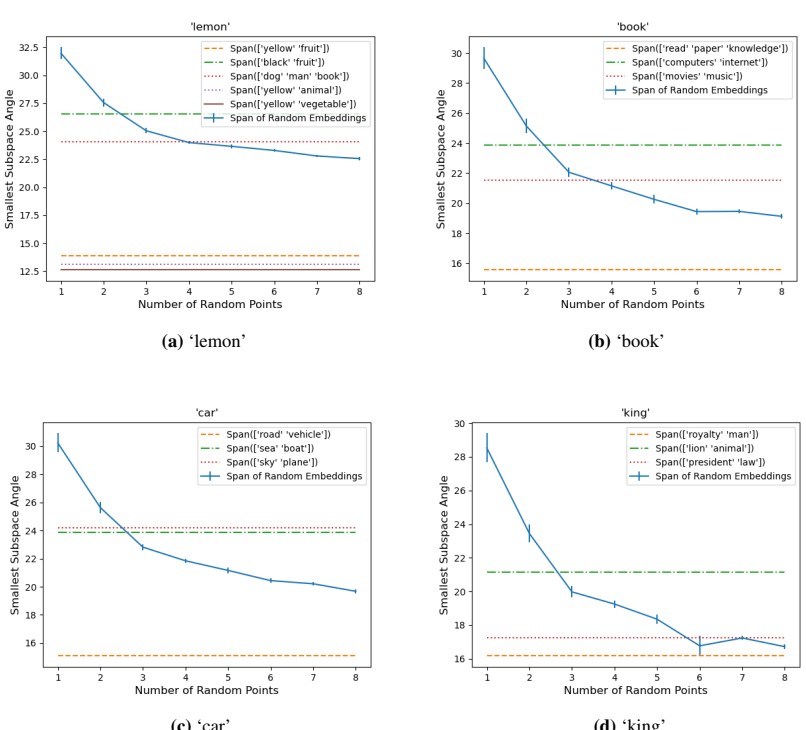

(a) 'lemon'

(b) 'book'

(c) 'car'

(d) 'king'

**Figure 4:** Linear subspaces spanned by estimated generalized Markov boundaries have smallest subspace angles with linear subspaces spanned by embeddings that best match semantic meanings

