# OpenReview forum: "Uncovering Meanings of Embeddings via Partial Orthogonality"
_NeurIPS.cc/2023/Conference — NeurIPS 2023 poster_

### Official Review · Reviewer_zRa2 · 2023-06-30

**Soundness:** 3 good
**Presentation:** 3 good
**Contribution:** 2 fair
**Rating:** 6
**Confidence:** 3

**Summary:**

This paper aims to uncover the semantic meaning of embedding vectors
within a given space. The basic idea is to determine
a generalized Markov boundary by computing the cosine similarity of the
orthogonality projected vectors within a subspace. The top K
candidates are then selected. Furthermore, the authors provide a theoretical
analysis of the concept of Independence preserving embedding in section
five.

**Strengths:**

- the connection between graphical model theory and explanation of embeddings seems novel (although similar ideas are present in other fields, such as the study of knowledge graph embeddings)
- conceptually, everything is well defined and formally presented
- The research question is clear and meaningful.
- The structure of this paper is well-organized. In particular, in section
two, the authors explain the necessary background information clearly

**Weaknesses:**

- experiments and results analysis is rather sparse, the paper has more focus on the theory and definitions
- experimental setup can be criticized (see comments below)
- related work with respect to knowledge graph embeddings could be more thorough. Several studies have focused on using projection or rotation techniques for KG embedding to predict the missing relationship between two entities [THW, SLH, SW]. Can these KG meth-
ods be adapted to uncover meaningful word embeddings?

[THW] Yun Tang, Jing Huang, Guangtao Wang, Xiaodong He, and Bowen Zhou. Orthogonal
relation transforms with graph context modeling for knowledge graph embedding. arXiv
preprint arXiv:1911.04910, 2019.
[SLH] Tengwei Song, Jie Luo, and Lei Huang. Rot-pro: Modeling transitivity by projection
in knowledge graph embedding. Advances in Neural Information Processing Systems,
34:24695–24706, 2021.
[SW] Baoxu Shi and Tim Weninger. Proje: Embedding projection for knowledge graph comple-
tion. In Proceedings of the AAAI Conference on Artificial Intelligence, volume 31, 2017.

**Questions:**

- What is the critical motivation for using Image-language pre-train model
CLIP as embedding? It would be better to compare it with other embedding techniques such as GloVe, word2vec, and BERT. And How does the
dimensionality of the embedding vectors decide the performance of the
proposed algorithm?
- This paper only provides five examples in Table 2 regarding meaningful
semantic evaluation. Would it be possible to compute the precise numer-
ical results using semantic evaluation metrics to show the advantage after
orthogonality projection?
- In section five, the author introduces the concept of IPE, which might
be more straightforward to understand through toy examples or visual
figures

---

> ### Author Rebuttal · Authors · 2023-08-10
>
> Thank you for your questions! We're glad you found the ideas novel and the paper well written.
>
> **Experiments**
> We completely agree with the reviewer that the paper is more focused on theory and definitions. The reason we consider pre-trained image-language model CLIP is that we found it tends to encode more interpretable semantic meanings. For example, for the target word “eggplant”, the word “purple” appears more meaningful in CLIP embedding than other text-based only embeddings.
>
>
> **Connection to knowledge graph**
> Thanks for suggesting these related works. We’ll include them in the updated version. On the other hand, although the “independence model” is a concept closely related to the graphical model, its abstract notion has applicability beyond graphs. In fact, many independence models cannot be embedded in graphs.
>
> **“This paper only provides five examples in Table 2 regarding meaningful semantic evaluation. Would it be possible to compute the precise numerical results using semantic evaluation metrics to show the advantage after orthogonality projection?”**
> Thanks for the suggestion! We have run additional experiments by using Wu-Palmer similarities as the numerical estimates. Please see the global review for more details.

---

> > ### Comment · Reviewer_zRa2 · 2023-08-22
> >
> > Thank you for your reply, it will be helpful in the discussion about the paper.

---

> ### Author Response · Authors · 2023-08-16
>
> Thank you again for your review and feedback. Do you have any additional concerns or questions? If you are satisfied with the response, we hope you will consider increasing the score.

---

### Official Review · Reviewer_DCJZ · 2023-07-02

**Soundness:** 2 fair
**Presentation:** 3 good
**Contribution:** 3 good
**Rating:** 4
**Confidence:** 3

**Summary:**

This paper investigates the relationship between the semantics and linear algebraic structure of token embeddings. It proposes utilizing partial orthogonality to define the "Markov boundary" of token embeddings. Given that token embeddings have limited dimensions and the Markov boundary can consist of numerous embeddings, the authors suggest relaxing the definition of partial orthogonality. They subsequently introduce an approximate algorithm to identify this boundary, which iteratively locates embeddings with high cosine similarity to the target vector after projecting onto orthogonal complement subspaces. To justify the effectiveness of vector space, the authors present the concept of independence preserving embeddings, which serves as the foundation for studying linear algebraic independence in embedding vectors.

**Strengths:**

1. The paper formally discussed the relationship between meanings of tokens and their algebraic independence.  It generalizes the idea of Markov boundary and relaxes its definition so it can be practically applied to word embeddings.

2. To validate the use of linear algebraic independence relationships between embeddings for studying their semantics, the author introduces the concept of independence preserving embedding. This concept demonstrates that embeddings maintain the independence structure of distribution, making the paper comprehensive and self-contained.

3. The authors conduct experiments using CLIP embeddings and demonstrate that their algorithm effectively identifies intriguing patterns between word embeddings, indicating that these embeddings possess semantic meanings.


**Weaknesses:**

1. Although the authors aim to study the independence relationship between word embeddings, they do not provide an evaluation metric to substantiate the effectiveness of the proposed method. The experimental results are presented as case studies with a limited number of words as examples, which may not be compelling for readers. A more robust experimental section would be beneficial.

2. The experiments conducted in the paper focus solely on the CLIP embedding model. While it is understandable that CLIP, being trained with visual information, may encode semantics that are more meaningful to humans, it would be interesting to explore whether the proposed method can be applied to other models that rely exclusively on text-based training.

**Questions:**

1. The embeddings studied in this paper is static embedding. I wonder what if the embeddings are contextualized? For example, what would happen if the embeddings are processed by Transformers?

2. Throughout the section 3, the definitions of d and n are confusing. It says if d is smaller or equal to n, vectors are linearly independent. Should it be the opposite? And the same things happens to line 132.

3. Typo: In line 96, period should be replaced by comma.

---

> ### Author Rebuttal · Authors · 2023-08-10
>
> Thank you for your thorough review and suggestions!
>
> **“Although the authors aim to study the independence relationship between word embeddings, they do not provide an evaluation metric to substantiate the effectiveness of the proposed method.”**
> The experiment section is divided into two parts. The first part tests the effectiveness of the proposed method in finding subspaces that can reduce residual correlations. The second part is the qualitative study, which evaluates whether the Markov boundary is semantically meaningful. In the paper, we measure semantic meaningfulness via human intuition. How to quantify this measure is an open question and we feel like deserves its own paper. Nevertheless, we include some additional experiments that have numerical evaluations of our method. Please see the top review for more details.
>
> **“The experiments conducted in the paper focus solely on the CLIP embedding model”**
> Thanks for the suggestion! And you’re right. We found that CLIP text embedding tends to encode more interpretable semantics meanings. For instance, for the word “eggplant”, the word “purple” appears more meaningful in CLIP embedding than other text-based only embeddings.
>
>
> **“The embeddings studied in this paper are static embedding. I wonder what if the embeddings are contextualized? For example, what would happen if the embeddings are processed by Transformers?”**
>
> Thanks for the suggestion! It would be an exciting future direction to extend this work to contextualized embeddings. On the other hand, in this paper, our experiments are done on nouns whose meanings are less ambiguous like “eggplant” and “zebra” to demonstrate the effectiveness of Markov boundaries.
>
> **“The definitions of d and n are confusing.”**
>
> Thanks for your careful attention and sorry for the confusion. It is a typo. We’ll correct it in the updated version.

---

> ### Author Response · Authors · 2023-08-16
>
> Thank you again for your review and feedback. Do you have any additional concerns or questions? If you are satisfied with the response, we hope you will consider increasing the score.

---

> ### Comment · Reviewer_DCJZ · 2023-08-17
>
> Thanks for the response and the new experiments. It addresses some of my questions. However, I believe a high-quality paper should be tested systematically on a large corpus with scientific metrics, instead of the case study. I would argue to keep my current score.

---

> > ### Author Response · Authors · 2023-08-18
> >
> > Thank you for your valuable feedback! We completely agree that testing the semantic relevance of learned Markov boundaries systematically with better metrics would be ideal. However, the dilemma we have is that, as far as we know, there are no good metrics in the literature. And we believe that large-scale evaluation with convincing metrics is an open problem and deserves its own work. Although there are existing metrics on semantic similarities that are based on Wordnet like Wu-Palmer similarity, they do not fit our experiments on Markov bounries. For example, the Wu-Palmer similarity score between “**eggplant**” and “**purple**” is only 0.167 but the score between “**eggplant**” and “**lemon**” is 0.667. However, because we want to construct the Markov boundary to be a minimal description set of the target word “**eggplant**”, one would expect to include “**purple**” instead of “**lemon**” despite what the WP score suggests. To see this, we asked ChatGPT to come up with a short description of the word “eggplant” and the answer is “a **purple** or dark-colored vegetable with a smooth skin, often used in cooking and known for its mild flavor.” This also fits human intuition.
> >
> > The previous example showcases the difficulty of coming up with good semantic metrics. Nevertheless, we humbly argue that the experiments in the paper, along with the newly added ones in the rebuttal, support the claims made in the paper. In particular, the central hypothesis of the paper is that partial orthogonality, and its byproduct Markov boundary, of embeddings, carries semantic information. To verify this claim, we provide both _quantitative_ and _qualitative_ experiments. For qualitative experiments, we appeal to human intuition by comparing the principal angle between learned Markov boundaries of the target embedding with linear subspaces spanned by relevant word embeddings of the target. For quantitative experiments, _unlike qualitative case studies_, we come up with a numerical estimate that calculates the average principal angle between learned Markov boundaries and embeddings of target descriptions from a dataset we created that consists of target words and their corresponding short descriptions.
> >
> > Although we do not claim the immediate practical impacts on a large scale, we believe that studying the Markov boundary of embeddings holds promise for understanding the inner workings of embeddings. This is one of our main scientific contributions and our empirical evaluations are designed to ascertain the utility of this claim. Because coming up with a good semantic metric is an open problem, we are of the opinion that it shouldn’t limit the contributions of our paper which is more focused on theories and definitions.

---

### Official Review · Reviewer_3K9o · 2023-07-06

**Soundness:** 1 poor
**Presentation:** 1 poor
**Contribution:** 2 fair
**Rating:** 2
**Confidence:** 4

**Summary:**

The central question (quoting the paper) is "How to make sense of an embedding vector in relation to other embedding vectors?"  For that
purpose, the authors propose to generalize the idea of the Markov boundary to embeddings, with a relaxed adaptation of this notion to
cope with word embeddings peculiarity. The paper then introduces an algorithm to find (approximately) what is called the generalized Markov boundary for a given embedding. Empirical evaluations are carried out on CLIP.

**Strengths:**

The real scientific goal should be first clarified before one can assess the strength of this proposition.


**Weaknesses:**

Maybe I completely misunderstand this paper, but I cannot tell what is its scientific goal. For me everything is confused in the paper: the notion of markov boundary for vectors in the context of contextualized embeddings (like CLIP), why so much formal definitions for at the end a rough relaxation.




**Questions:**

I have no question.

---

> ### Author Rebuttal · Authors · 2023-08-10
>
> Thanks for your review!
>
> **“The scientific goal”**
>
> We apologize for the confusion. In this paper, we hypothesize that semantic meanings have an independence structure. We use the abstract “independence model” to formalize this idea. On the other hand, embeddings, which are vector representations of words, should also have a similar independence structure. A natural candidate “independence model” for vector space is partial orthogonalities.
>
> _Therefore, the main conjecture of this paper is that partial orthogonalities of embedding spaces encode semantic meanings._ To test this theory, we first study the theoretical aspect of the problem. In particular, we generalize the notion of the Markov boundary to embeddings. The relaxation is necessary because the intersection property of “partial orthogonality” rarely holds for embeddings. And then we verify the theory empirically in our experiment section. Specifically, we examine whether the Markov boundary effectively conveys "semantic meanings."

---

> > ### Comment · Area_Chair_CD4j · 2023-08-21
> > **Acknowledgement**
> >
> > Dear authors,
> >
> > Thank you for your response here. You have clarified the scientific goal succinctly and I assure you we will take this answer into account in the upcoming discussion and decisions.
> >
> > best
> >
> > the ac

---

> > > ### Author Response · Authors · 2023-08-21
> > >
> > > Thanks for your reply!  We really appreciate your commitment to the quality of the review process.

---

> ### Author Response · Authors · 2023-08-16
>
> Thank you again for your review and feedback. Do you have any additional concerns or questions? If you are satisfied with the response, we hope you will consider increasing the score.

---

### Official Review · Reviewer_1qJm · 2023-07-07

**Soundness:** 3 good
**Presentation:** 2 fair
**Contribution:** 3 good
**Rating:** 6
**Confidence:** 2

**Summary:**

This paper presents some theory and a method for reasoning about information gain in embedding space via a relaxation of conditional independence, as well as some theory on independence preserving embeddings.

As information gain is inherently linked to independence, the paper focuses on defining a generalization of the Markov boundary that is meaningful in embedding space. The Markov boundary is a set of embeddings that "separate" the target embedding from all other test embeddings not in the boundary. Concretely, this means the cosine similarity between the projection of the target and test embeddings onto the orthogonal complement of the generalized Markov boundary should be 0. The proposed generalization relaxes elementwise orthogonality to distributional orthogonality, where the cosine similarities between the projections are allowed to cancel out, rather than all be 0. This criterion is motivated by practical concerns, where embeddings are low-dimensional representations where orthogonal residuals are rare.

Finding a generalized Markov boundary for a single target embedding is then accomplished by sampling a number of random sets of embeddings, finding the top $K$ embeddings that contain information about the target given the random embedding sets, then constructing the boundary from within those top $K$ embeddings.

Separately, the paper addresses another question of how to embed a set of independence assumptions in a lower dimensional space. A theorem is presented that shows that one can preserve independence assumptions as residual orthogonality to some degree, depending on the dimension.

Experiments are presented, using CLIP embeddings, that show that generalized Markov boundaries can indeed be found, that projecting onto the orthogonal complement of the span random embeddings is meaningful, and that the discovered generalized Markov boundaries are more aligned with the span of embeddings of related words than unrelated.

**Strengths:**

1. The first research question of reasoning about conditional independence and information gain in embedding space is appealing. Recent works in resolving ambiguity through dialogue, such as for image retrieval through 20 questions, reason over the space of individual images. However, if there are many images, this is not scalable. Intuitively, many of those images are likely to be very similar, motivating reasoning in the much lower dimensional CLIP embedding space.
2. The proposed definition of generalized Markov boundary and method for finding boundaries are reasonable.
3. The second research question about independence preserving embeddings is also worth studying for the same reason as above: potential applications would be very interesting.

**Weaknesses:**

1. The method and experiments for the generalized Markov boundary only involve a single target embedding.
1. The experimental evaluation only evaluates token embeddings, whereas the text encoder in CLIP can encode sequences. An experiment involving reasoning over sequence embeddings would make the paper much stronger, especially if the experiment involved a realistic task.
1. No experiments are performed for independence preserving embeddings.

**Questions:**

1. Primarily, I would like to see experiments verifying Theorem 13 on dimensionality reduction in independence preserving embeddings.
1. Is the only difference between kernel mean embeddings and independence preserving embeddings the choice of a kernel with finite dimensional feature map?
1. What is the relationship to work on information theory with kernel methods [2]?
1. A more realistic application that utilizes the method developed in the paper would greatly strengthen the paper. One possible application is an image retrieval game such as 20 questions [1], where the goal is to retrieve the correct image out of a set by asking questions.

[1] White, Julia, et al. "Open-domain clarification question generation without question examples." Proceedings of the 2021 Conference on Empirical Methods in Natural Language Processing. 2021.
[2] Francis Bach. Information Theory with Kernel Methods. 2022. ⟨hal-03577992v2⟩

**Limitations:**

I did not find a discussion of limitations.

---

> ### Author Rebuttal · Authors · 2023-08-10
>
> **“The method and experiments for the generalized Markov boundary only involve a single target embedding.”**
>
> The goal of this paper is to find good descriptions/explanations of given embeddings with other embedding vectors. We feel like it does not make too much sense to try to describe/explain multiple word embeddings at the same time. However, one could consider embeddings of phrases and sentences.
>
> **“The experimental evaluation only evaluates token embeddings, whereas the text encoder in CLIP can encode sequences.”**
>
> Thanks for the suggestion! Because our objective is to uncover the semantic meanings of words, finding the semantic meanings of sentences is a much harder task and very difficult to define, although one could use contextualized embeddings to deal with polysemantic words. In this paper, to avoid ambiguity, we choose to study nouns whose meanings are less ambiguous like “eggplant” and “zebra”.
>
> **“No experiments are performed for independence preserving embeddings.”**
>
> The study of IPE is mainly used to answer the theoretical question of whether such embeddings are possible. Its practical application is an interesting future research direction. Nevertheless, we include some additional experiments on IPE. Please see the top review for more details.
>
>
> **“Relation with kernel mean embedding and information theory with kernel method”**
>
> That’s a good question! We will try to explain this better in the updated version. In general, kernel embeddings are infinite dimensional embeddings that are more concerned with the moments of distributions they are trying to embed. For kernel mean embedding, it’s the first moment, and for information theory with kernel method, it’s the second moment. On the contrary, our independence-preserving embedding is a finite-dimensional embedding. Such finite dimension embedding can be directly used in practice. The reason we can produce finite dimension embedding is that we are only focused on the _structure_ of distributions, i.e., conditional independence statements, and not other statistical properties of the distributions.
>
> **“A more realistic application that utilizes the method developed in the paper would greatly strengthen the paper.”**
>
> Thanks for the suggestion! Finding an interesting application for our results is definitely an interesting next direction.

---

> ### Author Response · Authors · 2023-08-16
>
> Thank you again for your review and feedback. Do you have any additional concerns or questions? If you are satisfied with the response, we hope you will consider increasing the score.

---

> > ### Comment · Reviewer_1qJm · 2023-08-17
> >
> > In response to the rebuttal, and in hindsight, my original score is too harsh and will be raised from reject to weak accept with lower confidence. The paper supports its 3 contributions of generalized Markov embeddings, empirical validation, and independence preserving embeddings.
> >
> > My initial review was caused by a mismatch between the potential impact of the research question in this paper and the experimental validation presented. The paper presents theory towards embedding-based reasoning, and does not make a claim about large-scale empirical impact. The CLIP experiments in the paper are a small-scale study that validates the paper's claim. Further, larger-scale evaluation of the method is an opportunity for future work, and will not be held against the current paper.

---

> > > ### Author Response · Authors · 2023-08-18
> > >
> > > Thanks! We sincerely appreciate your time and effort to understand our work, and for updating your score accordingly.

---

### Author Rebuttal · Authors · 2023-08-10

We thank the reviewers for their thoughtful reviews and for recognizing the ideas presented in the paper as novel and appealing. In addition, we ran some quick experiments to additionally verify the effectiveness of our method and theory as suggested by reviewers. Tables and figures are included in the pdf file.

**Random projections**
In the paper, we show that top correlated words after random projections are more semantically meaningful by presenting some examples. To provide a numerical estimate as suggested by Reviewer zRa2, we adopt the Wu-Palmer similarity metric. We run the experiments in two settings. In the first setting, 1000 target words are randomly selected from the Brown corpus. As those words might contain rare and obscure ones, we run experiments in the second setting with 300 common nouns as target words. These words are selected by ChatGPT. Table 1 shows that the average WP similarities of top correlated words with target words are higher after random projections.

**Markov boundary**
In the paper, we demonstrate the effectiveness of our Markov boundary learning algorithm by measuring the smallest principal angle between Markov boundaries and random subspaces as well as some subspaces spanned by relevant words to target words. We present some selected examples in the paper. The additional experiments here provide numerical estimates. We first ask ChatGPT to give us a list of 50 common nouns each with a short description. Next, we convert the short descriptions into embeddings using CLIP text encoder. Finally, we compare the principal angle between description embeddings and subspaces spanned by Markov boundaries as well as randomly selected words. Figure 1 shows the learned Markvo boundaries consistently have smaller angles with description embeddings.

**IPE**
In the paper, we show the theories of independence-preserving embeddings. Here, we present additional empirical evidence of the theories. We randomly generate graphs using Erdős–Rényi model with p = 0.01 and the number of nodes to be 1000. We first apply the IPE construction method to get the embeddings and then apply random projections. Figure 2 shows that without projection (dimsion=1000), the average absolute cosine similarities of residuals after projecting onto respective Markov boundaries are nearly zero. As the random projection dimension increases, the average absolute cosine similarities increase slowly.

---

### Decision · Program_Chairs · 2023-09-21

**Decision:**

Accept (poster)

**Comment:**

This paper builds upon a vector space model of independence in order to propose a way of capturing the "meaning" of an embedding via a Markov boundary consisting of other embeddings.

The reviews mostly find the work interesting, but hard to evaluate; the authors have proposed some additional results that help. I agree that the mathematical model is interesting here and believe that this paper deserves some attention, and thus recommend acceptance. I stress however that the paper is rather weakly anchored in linguistic or lexicographic theory, approaching concepts such as "meaning" in a fairly rough way, somewhat conflated with similarity. I would also strongly recommend updating the additional experiments (submitted at rebuttal time) to use more reproducible sources for word definitions and word frequency such as dictionaries rather than ChatGPT.